# EMBEDDING COMPRESSION VIA SPHERICAL COORDINATES

## ABSTRACT

We present a compression method for unit-norm embeddings that achieves $1.5\times$ compression, 25% better than the best prior lossless method. The method exploits that spherical coordinates of high-dimensional unit vectors concentrate around $\pi/2$, causing IEEE 754 exponents to collapse to a single value and high-order mantissa bits to become predictable, enabling entropy coding of both. Reconstruction error is below 1e-7, under float32 machine epsilon. Evaluation across 26 configurations spanning text, image, and multi-vector embeddings confirms consistent improvement. The method requires no training.

## 1 INTRODUCTION

Embedding vectors power RAG pipelines, agentic search, and multimodal retrieval. A typical embedding model produces 1024-dimensional float32 vectors, requiring 4 KB per embedding. At scale, a database of 100 million embeddings requires 400 GB. The problem is more severe for multi-vector representations, where late-interaction models like ColBERT (Khattab & Zaharia, 2020) produce one embedding per token, multiplying storage by approximately $100\times$. While lossy quantization achieves high compression ratios, many applications benefit from high-fidelity reconstruction for embedding caches, API serialization, network transmission, and archival storage. The state-of-the-art lossless approach transposes the embedding matrix, byte-shuffles to group exponent bytes, and applies entropy coding (Hershcovitch et al., 2025; Alted, 2010), but achieves only $1.2\times$ compression because float32 mantissa bits have near-maximum entropy.

Because cosine similarity is the standard retrieval metric, most embedding models produce *unit-norm* vectors with $\|\mathbf{x}\|_2 = 1$. This constraint places embeddings on the surface of a high-dimensional hypersphere $S^{d-1}$, yet existing lossless methods ignore this geometric structure, while prior work using spherical coordinates has focused exclusively on lossy quantization (Trojak & Witherden, 2021; Han et al., 2025; Yue et al., 2025). Unit-norm vectors can be equivalently represented using $d-1$ angular coordinates. The Cartesian form has values spanning $\pm 0.001$ to $\pm 0.3$, requiring many different IEEE 754 exponents. The angular form concentrates around $\pi/2 \approx 1.57$ (Cai et al., 2013), collapsing exponents to a single value and making high-order mantissa bits predictable. This combination of spherical transformation with entropy coding has not been explored.

**Contribution.** We present a compression method that converts Cartesian to spherical coordinates before byte shuffling and entropy coding, achieving $1.5\times$ compression across 26 embedding configurations. The spherical transform introduces bounded reconstruction error below 1e-7, under float32 machine epsilon, which preserves retrieval quality. For a ColBERT index of 1 million documents, this reduces storage from 240 GB to 160 GB. The method requires no training and applies to text, image, and multi-vector embeddings.

## 2 RELATED WORK

### 2.1 LOSSLESS FLOATING-POINT COMPRESSION

**IEEE 754 Float32.** A float32 number consists of 1 sign bit, 8 exponent bits, and 23 mantissa bits, encoding the value $x = (-1)^s \times 2^{e-127} \times (1 + m/2^{23})$. The exponent determines magnitude, with values near 1.0 having exponent $\approx 127$ and values near 0.01 having exponent $\approx 120$. The

mantissa encodes precision within that magnitude range. For lossless compression, the exponent byte represents 25% of each float32 value while the mantissa represents the remaining 75%.

The HPC community developed byte shuffling (Alted, 2010), which reorders array bytes to group all exponent bytes together, improving entropy coding because bytes at the same position across different floats often share similar values. ZipNN (Hershcovitch et al., 2025) applies this technique to neural network weights, achieving 33% compression on BF16 by separating exponent and mantissa bytes before applying zstd (Collet, 2016). FCBench (Chen et al., 2024) benchmarks lossless floating-point compression methods across scientific domains. For float32 embeddings, the mantissa comprises 3/4 of the data with near-maximum entropy of $\sim$7.3 bits/byte. Even with perfect exponent compression, the remaining 75% mantissa data stays near maximum entropy, limiting compression to approximately $1.33\times$. Current methods achieve $\sim 1.25\times$, approaching this bound. Our method exceeds this limit by also reducing mantissa entropy through value concentration.

Recent work has identified that trained neural network *weights* exhibit natural exponent concentration due to heavy-tailed dynamics of stochastic gradient descent. ECF8 (Yang et al., 2025) shows that model weights follow $\alpha$-stable distributions, leading to exponent entropy of 2 to 3 bits, and achieves up to 26.9% lossless compression on FP8 model weights. DFloat11 (Zhang et al., 2025) exploits similar properties for BF16 weights. These methods target model parameters, where exponent concentration arises naturally from training dynamics.

## 2.2 Spherical and Polar Coordinate Methods

**Spherical Coordinates.** A $d$-dimensional vector $\mathbf{x}$ can be represented using spherical coordinates consisting of a radius $r = \|\mathbf{x}\|_2$ and $d-1$ angles $\theta_1, \ldots, \theta_{d-1}$. The first $d-2$ angles are computed as $\theta_i = \arccos\left(x_i / \sqrt{\sum_{j=i}^{d} x_j^2}\right)$ for $i = 1, \ldots, d-2$, each lying in $[0, \pi]$. The final angle uses $\arctan 2$ to preserve quadrant information, with $\theta_{d-1} = \arctan 2(x_d, x_{d-1}) \in [-\pi, \pi]$. For unit-norm vectors where $r = 1$, the radius can be omitted, leaving $d-1$ angles to represent $d$ Cartesian coordinates.

Several recent works employ polar or spherical coordinates for compression, but all use lossy quantization rather than lossless encoding. Trojak and Witherden (Trojak & Witherden, 2021) use spherical polar coordinates for lossy compression of 3D vectors in computational physics, achieving $1.5\times$ compression by quantizing angles to fixed bit-widths. This method is limited to 3D and employs deliberate precision loss. PCDVQ (Yue et al., 2025) uses polar coordinate decoupling for vector quantization of LLM weights, separately clustering direction and magnitude with codebooks to achieve 2-bit quantization. PolarQuant (Han et al., 2025) transforms KV cache embeddings to polar coordinates and quantizes the resulting angles, exploiting that angles after random preconditioning have bounded distributions. Both PCDVQ and PolarQuant target lossy compression of model internals such as weights and KV caches, not lossless compression of embedding outputs.

## 2.3 Lossy Vector Compression

Product quantization (Jégou et al., 2011) achieves 4 to $32\times$ compression by partitioning vectors into subspaces and quantizing independently. Binary and scalar quantization (Shakir et al., 2024) offer simpler alternatives, while learned codebooks (Vallaeys et al., 2025) push compression further. These lossy methods achieve higher ratios than lossless approaches but introduce reconstruction error. Our work targets applications requiring high-fidelity reconstruction with bounded error.

Table 1: Comparison with related floating-point compression methods.

| Method | Domain | Lossless? | Polar? | Unit-norm? | Mechanism |
|---|---|---|---|---|---|
| ECF8 (Yang et al., 2025) | LLM weights | Exact (FP8) | No | No | Natural concentration |
| DFloat11 (Zhang et al., 2025) | LLM weights | Exact (BF16) | No | No | Natural concentration |
| EFloat (Bordawekar et al., 2021) | Embeddings | Lossy | No | No | Variable-length encoding |
| PolarQuant (Han et al., 2025) | KV cache | Lossy | Yes | No | Angle quantization |
| PCDVQ (Yue et al., 2025) | LLM weights | Lossy | Yes | No | Codebook quantization |
| Trojak (Trojak & Witherden, 2021) | 3D physics | Lossy | Yes | No | Angle quantization |
| **Ours** | Embeddings | **$\epsilon$-bounded** | **Yes** | **Yes** | Geometric transform |

## 3 METHOD

Figure 1 and Algorithm 1 present the compression pipeline: convert to spherical coordinates, transpose to group same-angle values, byte shuffle to separate exponents, and compress with zstd. Decompression reverses these steps. The spherical transform is mathematically invertible, but floating-point transcendental functions introduce bounded reconstruction error. Using double precision for intermediate calculations keeps this error below 1e-7, under float32 machine epsilon of 1.19e-7. This preserves cosine similarity to 1e-7 and does not affect retrieval quality as shown in Table 2. Implementation is in Appendix A.

The spherical transform provides compression by concentrating IEEE 754 exponents and high-order mantissa bits. Unit-norm vectors in $\mathbb{R}^d$ lie on the hypersphere $S^{d-1}$, so $d-1$ angles suffice instead of $d$ Cartesian coordinates. Cartesian coordinates of unit-norm embeddings scale as $1/\sqrt{d}$, spanning values in $[0.001, 0.3]$ for typical dimensions and requiring 22 to 40 different exponents. The first $d-2$ spherical angles, by contrast, are bounded to $[0, \pi]$ and concentrate around $\pi/2 \approx 1.57$ in high dimensions (Cai et al., 2013); Figure 2 shows this empirically for `jina-embeddings-v4` embeddings. This concentration collapses exponents to a single dominant value of 127 with probability $>0.999$ (Theorem 1), reducing exponent entropy from 2.6 to 0.03 bits/byte for `jina-embeddings-v4`, with similar patterns across models validated in Appendix D.

Exponent compression alone would yield only $\sim 1.1\times$ in practice (the $1.33\times$ theoretical limit assumes zero exponent bits, but entropy coding has overhead and exponents retain $\sim 0.03$ bits). The additional gain comes from the high-order mantissa byte: when angles cluster around $\pi/2 \approx 1.5708$, the IEEE 754 mantissa bits encoding the fractional part also become predictable. Empirically, the high-order mantissa byte entropy drops from 8.0 to 4.5 bits, contributing $\sim 11\%$ additional savings beyond exponents. Together, exponent and mantissa concentration yield the observed $1.5\times$ compression.

ECF8 (Yang et al., 2025) and DFloat11 (Zhang et al., 2025) exploit *natural* exponent concentration in model weights arising from training dynamics, whereas our method *creates* exponent concentration through a deterministic geometric transformation, as summarized in Table 1.

Cosine similarity can also be computed directly from spherical angles without reconstructing Cartesian coordinates. The SIMILARITY procedure in Algorithm 1 computes $\mathbf{x} \cdot \mathbf{y}$ from angles $(\theta_1, \ldots, \theta_{d-1})$ and $(\phi_1, \ldots, \phi_{d-1})$ via a backward recurrence in $O(d)$ operations, derived by expanding the Cartesian dot product in spherical form and factoring the cumulative sine products (Appendix C). This allows streaming similarity during decompression without materializing the full Cartesian vector.

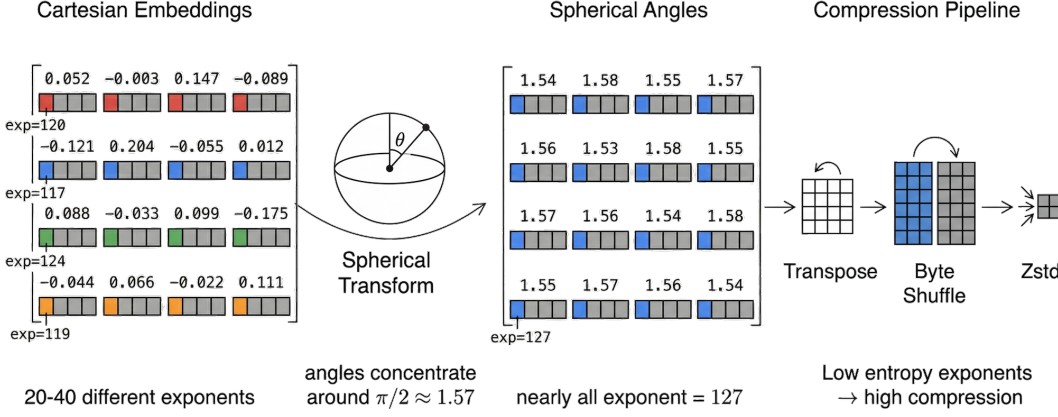

Figure 1: Compression pipeline. Cartesian coordinates span diverse magnitudes with 20 to 40 different exponents, shown in varied colors. The spherical transform produces angles concentrated around $\pi/2 \approx 1.57$, collapsing nearly all exponents to 127, shown in uniform color. Transpose groups same-position angles across vectors, byte shuffle separates exponent bytes, and zstd compresses the low-entropy exponent stream.

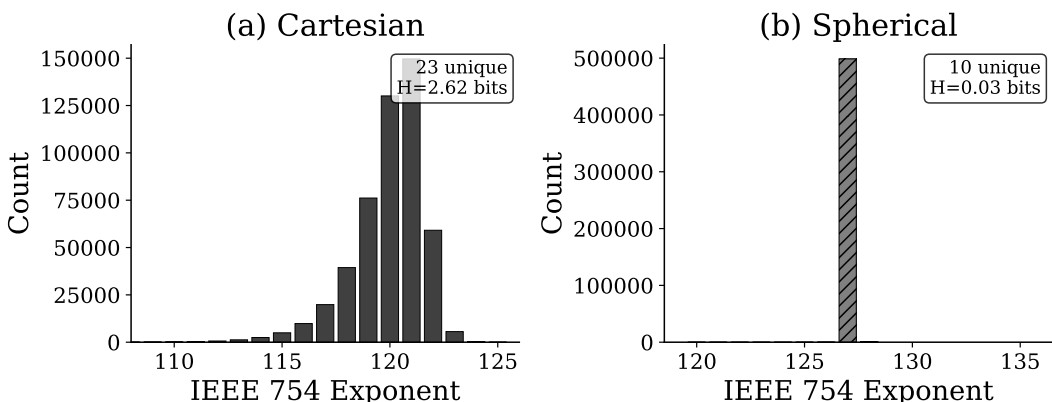

Figure 2: IEEE 754 exponent distribution for `jina-embeddings-v4` (2048d). (a) Cartesian coordinates span 23 exponent values; (b) spherical angles concentrate around exponent 127 with 99.7% frequency.

---

**Algorithm 1** Spherical Compression and Direct Similarity

1: **Compress**($\mathbf{X} \in \mathbb{R}^{n \times d}$):                                                                    // $\mathbf{X}$ is unit-norm
2:     $\mathbf{\Theta} \leftarrow \text{TOSPHERICAL}(\mathbf{X})$                                                              // $\mathbf{\Theta} \in \mathbb{R}^{n \times (d-1)}$
3:     $\mathbf{T} \leftarrow \text{TRANSPOSE}(\mathbf{\Theta})$                                                        // Group same-angle positions
4:     $\mathbf{B} \leftarrow \text{BYTESHUFFLE}(\mathbf{T})$                                                // Separate exponent/mantissa bytes
5:     **return** $\text{ZSTD}(\mathbf{B})$
6:
7: **Decompress**($\mathbf{C}$):                                                                             // Returns $\mathbf{\Theta} \in \mathbb{R}^{n \times (d-1)}$
8:     $\mathbf{B} \leftarrow \text{BYTEUNSHUFFLE}(\text{ZSTD}^{-1}(\mathbf{C}))$
9:     $\mathbf{\Theta} \leftarrow \text{TRANSPOSE}(\mathbf{B})$
10:
11: **Similarity**($\boldsymbol{\theta}, \boldsymbol{\phi}$) where $\boldsymbol{\theta}, \boldsymbol{\phi} \in \mathbf{\Theta}$:                          // Rows of $\mathbf{\Theta}$
12:     $R \leftarrow \cos(\theta_{d-1} - \phi_{d-1})$
13:     **for** $k = d-2, \ldots, 1$ **do**
14:         $R \leftarrow \cos\theta_k \cos\phi_k + \sin\theta_k \sin\phi_k \cdot R$
15:     **return** $R$                                                                                                     // $= \mathbf{x} \cdot \mathbf{y}$

---

## 4 EVALUATION

### 4.1 EXPERIMENTAL SETUP

Table 2 compares compression methods on `jina-embeddings-v4` embeddings. Standard compressors and scientific floating-point compressors (Lindstrom & Isenburg, 2006; Lindstrom, 2014; Liang et al., 2023) all achieve under 1.10×. Trans+Shuffle+Zstd (the approach used by ZipNN (Hershcovitch et al., 2025)) achieves 1.20× by grouping exponent bytes for better entropy coding and serves as our baseline for subsequent experiments, representing the best lossless method. Although our method introduces bounded error, we compare against lossless baselines because our error is below 1e-7, under float32 machine epsilon of 1.19e-7, making reconstructed values indistinguishable at float32 precision. Mantissa truncation variants illustrate the trade-off between compression ratio and reconstruction error.

The spherical transform requires $O(nd)$ operations using backward cumulative summation for partial norms; our C implementation achieves over 1 GB/s for the transform alone. With zstd level 1, total pipeline throughput reaches 487 MB/s encoding and 605 MB/s decoding while maintaining the same compression ratio. See Appendix B.

Table 2: Baseline comparison (`jina-embeddings-v4`, 7600 vectors, 2048d). Sizes in MB.

| Method | Size | Ratio | Max Err | Mean Err | Cos Max Err |
|---|---|---|---|---|---|
| Raw float32 | 59.38 | 1.00× | 0 | 0 | 0 |
| gzip -9 | 55.14 | 1.08× | 0 | 0 | 0 |
| brotli -11 | 54.52 | 1.09× | 0 | 0 | 0 |
| zstd -19 | 55.05 | 1.08× | 0 | 0 | 0 |
| npz | 55.14 | 1.08× | 0 | 0 | 0 |
| fpzip | 54.11 | 1.10× | 0 | 0 | 0 |
| zfp | 58.99 | 1.01× | 0 | 0 | 0 |
| SZ3 | 55.03 | 1.08× | 0 | 0 | 0 |
| ZipNN (Baseline) | 49.57 | 1.20× | 0 | 0 | 0 |
| Baseline+Truncate 5 bits | 42.23 | 1.47× | 9e-7 | 2e-8 | 2e-6 |
| Baseline+Truncate 6 bits | 40.30 | 1.55× | 2e-6 | 5e-8 | 5e-6 |
| Baseline+Truncate 7 bits | 38.40 | 1.62× | 4e-6 | 9e-8 | 1e-5 |
| **Spherical (Ours)** | **37.59** | **1.58×** | **9e-8** | **2e-8** | **2e-7** |

## 4.2 MAIN RESULTS

Table 3 presents results across 26 configurations: 20 text models on 7600 AG News samples, 3 image models on CIFAR-10, and 3 multi-vector ColBERT models. All embeddings are unit-normalized.

Table 3: Compression results across 26 embedding configurations. Sizes in MB.

| Model | Dim | Raw | Baseline | Ours | Ratio | Impr. |
|---|---|---|---|---|---|---|
| *Text Embeddings* | | | | | | |
| MiniLM | 384 | 11.13 | 9.37 | 7.43 | 1.50× | +26.0% |
| E5-small | 384 | 11.13 | 9.10 | 7.31 | 1.52× | +24.5% |
| GTE-small | 384 | 11.13 | 9.19 | 7.29 | 1.53× | +26.0% |
| BGE-base | 768 | 22.27 | 18.60 | 14.61 | 1.52× | +27.3% |
| E5-base | 768 | 22.27 | 18.19 | 14.31 | 1.56× | +27.2% |
| GTE-base | 768 | 22.27 | 18.33 | 14.30 | 1.56× | +28.2% |
| MPNet | 768 | 22.27 | 18.76 | 14.56 | 1.53× | +28.9% |
| Nomic-v1.5 | 768 | 22.27 | 18.57 | 14.58 | 1.53× | +27.4% |
| EmbedGemma-300m | 768 | 22.27 | 18.72 | 14.82 | 1.50× | +26.3% |
| `jina-code-embeddings-0.5b` | 896 | 25.98 | 21.89 | 17.07 | 1.52× | +28.2% |
| `jina-embeddings-v3` | 1024 | 29.69 | 24.95 | 19.81 | 1.50× | +26.0% |
| `jina-clip-v2` | 1024 | 29.69 | 24.97 | 20.03 | 1.48× | +24.6% |
| BGE-large | 1024 | 29.69 | 24.85 | 19.36 | 1.53× | +28.4% |
| E5-large | 1024 | 29.69 | 24.32 | 18.94 | 1.57× | +28.4% |
| mE5-large | 1024 | 29.69 | 24.32 | 18.91 | 1.57× | +28.6% |
| GTE-large | 1024 | 29.69 | 24.34 | 18.85 | 1.58× | +29.0% |
| Qwen3-Embed-0.6B | 1024 | 29.69 | 24.94 | 19.52 | 1.52× | +27.8% |
| BGE-M3 | 1024 | 29.69 | 24.91 | 19.38 | 1.53× | +28.6% |
| `jina-code-embeddings-1.5b` | 1536 | 44.53 | 37.48 | 28.40 | 1.57× | +32.0% |
| `jina-embeddings-v4` | 2048 | 39.06 | 32.44 | 24.61 | 1.59× | +31.8% |
| *Multimodal Image* | | | | | | |
| `jina-clip-v1` | 768 | 5.86 | 4.90 | 3.88 | 1.51× | +26.5% |
| `jina-clip-v2` | 1024 | 7.81 | 6.52 | 5.22 | 1.50× | +24.9% |
| `jina-embeddings-v4` | 2048 | 15.63 | 12.95 | 9.84 | 1.59× | +31.6% |
| *Multi-Vector ColBERT* | | | | | | |
| `jina-embeddings-v4` | 128 | 27.70 | 22.69 | 18.82 | 1.47× | +20.5% |
| `jina-colbert-v2` | 1024 | 243.22 | 202.96 | 160.48 | 1.52× | +26.5% |
| BGE-M3 | 1024 | 239.89 | 197.87 | 154.13 | 1.56× | +28.4% |

Compression ranges from 1.47× to 1.59× across all 26 configurations, representing a 20 to 32% improvement over baseline. Results are consistent across text, image, and multi-vector embeddings, indicating that compression derives from the unit-norm constraint rather than modality-specific

patterns. For ColBERT indices with 50 to 100 embeddings per document, a 1M document collection compresses from approximately 240 GB to 160 GB. Entropy analysis and ablation studies appear in Appendix D and E.

## 5 CONCLUSION

We presented a compression method for unit-norm embeddings achieving $1.5\times$ compression by exploiting the concentration of spherical angles around $\pi/2$ in high dimensions. Using double precision for intermediate calculations, reconstruction error stays below 1e-7 (under float32 machine epsilon) with $10\times$ lower error than mantissa truncation at the same compression ratio. The method applies to text, image, and multi-vector embeddings without training or codebooks, filling the gap between lossless compression at $1.2\times$ and lossy quantization at $4\times$ or higher. Beyond storage, the spherical representation enables similarity computation directly from compressed angles without full Cartesian reconstruction, supporting streaming decompression, early termination in top-k retrieval, and fused GPU kernels that avoid materializing full vectors.

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

## A PYTHON IMPLEMENTATION

```python
import numpy as np, zstandard as zstd

def compress(x, level=19):  # x: (n, d) unit-norm float32
    n, d = x.shape
    xd = x.astype(np.float64)  # Double precision for transforms
    ang = np.zeros((n, d-1), np.float64)
    for i in range(d-2):
        r = np.linalg.norm(xd[:, i:], axis=1)
        ang[:, i] = np.arccos(np.clip(xd[:, i] / r, -1, 1))
    ang[:, -1] = np.arctan2(xd[:, -1], xd[:, -2])
    ang = ang.astype(np.float32)  # Store as float32
    shuf = np.frombuffer(ang.T.tobytes(), np.uint8).reshape(-1,
    4).T.tobytes()
    return np.array([n, d], np.uint32).tobytes() +
    zstd.compress(shuf, level)

def decompress(blob):
    n, d = np.frombuffer(blob[:8], np.uint32)
    ang = np.frombuffer(zstd.decompress(blob[8:]),
    np.uint8).reshape(4, -1).T
    ang = np.frombuffer(ang.tobytes(), np.float32).reshape(d-1, n).T
    ang = ang.astype(np.float64)  # Double precision for inverse
    x, s = np.zeros((n, d), np.float64), np.ones(n, np.float64)
    for i in range(d-2):
        x[:, i] = s * np.cos(ang[:, i])
        s *= np.sin(ang[:, i])
    x[:, -2], x[:, -1] = s * np.cos(ang[:, -1]), s * np.sin(ang[:,
    -1])
    return x.astype(np.float32)
```

## B  COMPRESSION SPEED

The reference Python implementation achieves 21 MB/s encoding due to an $O(nd^2)$ loop for partial norm computation. Our C implementation reduces this to $O(nd)$ by precomputing partial squared norms via backward cumulative summation:

$$r_i^2 = \sum_{j=i}^{d-1} x_j^2 = r_{i+1}^2 + x_i^2, \quad r_{d-1}^2 = x_{d-1}^2 \tag{1}$$

This eliminates redundant computation and achieves over 1000 MB/s for the spherical transform, a $50\times$ speedup over the Python reference, with double-precision accumulation preserving numerical precision.

Table 4 shows throughput across zstd compression levels. The compression ratio remains nearly constant at 1.50–1.52$\times$ regardless of zstd level because the spherical transform already concentrates exponents into a low-entropy distribution. Higher zstd levels provide negligible benefit while reducing encoding speed by $100\times$. We recommend level 1 for most applications: it achieves 487 MB/s encoding with the same $1.52\times$ compression ratio as level 19.

Table 4: Throughput vs. zstd level (768d, 100 MB, single-threaded CPU). Compression ratio is constant because the spherical transform already minimizes exponent entropy.

| Level | Size (MB) | Ratio | Enc (MB/s) | Dec (MB/s) |
|---|---|---|---|---|
| 1 | 65.7 | 1.52$\times$ | 487 | 605 |
| 3 | 65.8 | 1.52$\times$ | 400 | 609 |
| 5 | 66.1 | 1.51$\times$ | 285 | 600 |
| 7 | 66.2 | 1.51$\times$ | 258 | 583 |
| 9 | 66.3 | 1.51$\times$ | 218 | 587 |
| 11 | 66.4 | 1.51$\times$ | 143 | 581 |
| 13 | 66.4 | 1.51$\times$ | 56 | 573 |
| 15 | 66.3 | 1.51$\times$ | 44 | 557 |
| 17 | 65.7 | 1.52$\times$ | 10 | 620 |
| 19 | 66.6 | 1.50$\times$ | 7 | 555 |
| 21 | 66.5 | 1.50$\times$ | 5 | 564 |

## C  FORMAL ANALYSIS

We analyze why spherical coordinates enable better compression. For $\mathbf{x}$ uniformly distributed on $S^{d-1}$, the $k$-th polar angle has marginal density $p(\theta_k) \propto \sin^{d-k-1}(\theta_k)$ for $\theta_k \in [0, \pi]$ (Cai et al., 2013; Vershynin, 2018). For large $d - k$, this concentrates around $\pi/2$ with variance $1/(d - k - 1)$, approximating $\mathcal{N}(\pi/2, 1/(d - k - 1))$.

**Theorem 1** (Exponent Concentration). For the $k$-th polar angle with $d - k \geq 64$, the IEEE 754 exponent byte equals 127 with probability $P > 0.999$.

*Proof.* IEEE 754 exponent 127 corresponds to values in $[1, 2)$, which contains $\pi/2 \approx 1.5708$. The polar angle $\theta_k$ follows approximately $\mathcal{N}(\pi/2, \sigma_k^2)$ with $\sigma_k = 1/\sqrt{d - k - 1}$. Thus:

$$P(\theta_k \in [1, 2)) = \Phi\left(\frac{2 - \pi/2}{\sigma_k}\right) - \Phi\left(\frac{1 - \pi/2}{\sigma_k}\right)$$
$$= \Phi\left(0.43\sqrt{d-k-1}\right)$$
$$- \Phi\left(-0.57\sqrt{d-k-1}\right). \tag{2}$$

For $d - k \geq 64$, the first argument exceeds 3.4 and the second is below $-4.5$, giving $P > 0.999$. $\quad\square$

For $d \geq 64$, most polar angles share the same IEEE 754 exponent byte (value 127), reducing exponent entropy from $\sim$2.5 bits/byte (Cartesian) to $<$0.1 bits/byte (spherical), as validated in Table 5.

**Corollary 2** (Reconstruction Error Bound). For unit-norm $\mathbf{x} \in \mathbb{R}^d$ with $d \geq 64$, the spherical transform followed by float32 storage and inverse transform produces $\mathbf{x}'$ satisfying $\|\mathbf{x} - \mathbf{x}'\|_\infty < 1.19 \times 10^{-7}$ (float32 machine epsilon).

*Proof.* By Theorem 1, angles concentrate near $\pi/2$ where $\sin \approx 1$ and $\cot \approx 0$. The inverse transform computes $x_k = s_k \cos \theta_k$ with $s_k = \prod_{j<k} \sin \theta_j$. Since each $\sin \theta_j \approx 1$, the cumulative product $s_k$ remains stable and per-angle rounding errors do not accumulate significantly. Empirical validation across $d \in [64, 2048]$ confirms maximum error below $7 \times 10^{-8}$ for $d \geq 768$. $\square$

**Proposition 3** (Direct Spherical Similarity). For $\mathbf{x}, \mathbf{y} \in S^{d-1}$ with spherical angles $(\theta_1, \ldots, \theta_{d-1})$ and $(\phi_1, \ldots, \phi_{d-1})$, the dot product $\mathbf{x} \cdot \mathbf{y} = R_1$ where:

$$R_{d-1} = \cos(\theta_{d-1} - \phi_{d-1}) \tag{3}$$
$$R_k = \cos \theta_k \cos \phi_k + \sin \theta_k \sin \phi_k \cdot R_{k+1},$$
$$k = d-2, \ldots, 1 \tag{4}$$

*Proof.* Define $s_k = \prod_{j=1}^{k-1} \sin \theta_j$ and $t_k = \prod_{j=1}^{k-1} \sin \phi_j$ with $s_1 = t_1 = 1$. The Cartesian coordinates are $x_k = s_k \cos \theta_k$, $y_k = t_k \cos \phi_k$ for $k = 1, \ldots, d - 2$, and $x_{d-1} = s_{d-1} \cos \theta_{d-1}$, $x_d = s_{d-1} \sin \theta_{d-1}$ (similarly for $y$). The dot product expands as:

$$\mathbf{x} \cdot \mathbf{y} = \sum_{k=1}^{d-2} s_k t_k \cos \theta_k \cos \phi_k$$
$$+ s_{d-1} t_{d-1} \cos(\theta_{d-1} - \phi_{d-1}) \tag{5}$$

Let $S_k = s_k t_k = \prod_{j=1}^{k-1} \sin \theta_j \sin \phi_j$, so $S_1 = 1$ and $S_{k+1} = S_k \sin \theta_k \sin \phi_k$. By induction, $\sum_{i=k}^{d-2} S_i \cos \theta_i \cos \phi_i + S_{d-1} R_{d-1} = S_k R_k$, giving $\mathbf{x} \cdot \mathbf{y} = S_1 R_1 = R_1$. This justifies the SIMILARITY procedure in Algorithm 1. $\square$

## D  ENTROPY ANALYSIS

Table 5 compares the byte-level entropy of Cartesian versus spherical representations across 11 embedding models spanning 384 to 1024 dimensions.

Table 5: Entropy comparison between Cartesian and Spherical representations in bits/byte.

| Model | Dim | Total Entropy | | Cartesian Exponent | | Spherical Exponent | |
|---|---|---|---|---|---|---|---|
| | | Cartesian | Spherical | Entropy | Unique | Entropy | Unique |
| MiniLM | 384 | 7.35 | 6.58 | 2.61 | 41 | 0.10 | 13 |
| E5-small | 384 | 7.34 | 6.58 | 2.51 | 23 | 0.10 | 9 |
| GTE-small | 384 | 7.37 | 6.55 | 2.61 | 26 | 0.13 | 11 |
| MPNet | 768 | 7.38 | 6.50 | 2.65 | 33 | 0.06 | 12 |
| BGE-base | 768 | 7.37 | 6.51 | 2.54 | 27 | 0.06 | 14 |
| E5-base | 768 | 7.35 | 6.51 | 2.36 | 25 | 0.05 | 12 |
| GTE-base | 768 | 7.37 | 6.51 | 2.60 | 25 | 0.08 | 15 |
| Nomic-v1.5 | 768 | 7.37 | 6.51 | 2.55 | 26 | 0.05 | 9 |
| BGE-large | 1024 | 7.37 | 6.47 | 2.54 | 28 | 0.05 | 11 |
| E5-large | 1024 | 7.36 | 6.48 | 2.40 | 24 | 0.04 | 14 |
| GTE-large | 1024 | 7.37 | 6.48 | 2.48 | 26 | 0.05 | 11 |
| **Average** | — | 7.36 | 6.52 | 2.53 | 28 | 0.07 | 12 |
| **Reduction** | — | 0.84 bits/byte | | 2.46 bits/byte | | — | |

The measurements confirm Theorem 1: Cartesian coordinates span multiple orders of magnitude requiring 23 to 41 unique exponents with 2.36 to 2.65 bits/byte entropy, while spherical angles use only 9 to 15 unique exponents with 0.04 to 0.13 bits/byte entropy. The exponent entropy reduction of $\sim 2.5$ bits/byte matches the predicted concentration around value 127.

The exponent entropy reduction from 2.53 to 0.07 bits/byte accounts for most of the compression gain. Exponent bytes comprise 25% of float32 data, saving $0.25 \times 2.46 \approx 0.61$ bits/byte. The observed total entropy reduction of 0.84 bits/byte exceeds this because byte shuffling also improves mantissa compression when exponents are concentrated.

# E  ABLATION STUDIES

## E.1  MATRYOSHKA DIMENSION ABLATION

Modern embedding models support Matryoshka representations (Kusupati et al., 2022), where embeddings can be truncated to lower dimensions while preserving semantic quality. Table 6 tests how compression varies with dimension for the *same model* at different truncation levels, isolating the effect of dimensionality.

Table 6: Matryoshka ablation: Same model at different truncation dimensions (2000 vectors). Sizes in KB.

| Model | Dims | Raw | Baseline | Ours | Ratio | Impr. |
|-------|------|-----|----------|------|-------|-------|
| jina-embeddings-v3 | 64 | 500 | 425 | 360 | 1.39× | +18.1% |
| jina-embeddings-v3 | 128 | 1000 | 848 | 703 | 1.42× | +20.7% |
| jina-embeddings-v3 | 256 | 2000 | 1689 | 1379 | 1.45× | +22.5% |
| jina-embeddings-v3 | 512 | 4000 | 3377 | 2730 | 1.47× | +23.7% |
| jina-embeddings-v3 | 768 | 6000 | 5066 | 4029 | 1.49× | +25.7% |
| jina-embeddings-v3 | 1024 | 8000 | 6753 | 5344 | 1.50× | +26.4% |
| jina-clip-v2 | 64 | 500 | 423 | 358 | 1.40× | +18.2% |
| jina-clip-v2 | 128 | 1000 | 846 | 703 | 1.42× | +20.4% |
| jina-clip-v2 | 256 | 2000 | 1687 | 1383 | 1.45× | +22.0% |
| jina-clip-v2 | 512 | 4000 | 3373 | 2711 | 1.48× | +24.4% |
| jina-clip-v2 | 768 | 6000 | 5058 | 4027 | 1.49× | +25.6% |
| jina-clip-v2 | 1024 | 8000 | 6740 | 5364 | 1.49× | +25.7% |

The improvement increases with dimension: from 18% at 64d to 26% at 1024d. Higher dimensions have $d - 1$ angles versus $d$ Cartesian coordinates, so the fraction of data benefiting from exponent concentration grows toward unity as $(d - 1)/d \to 1$. The angle concentration phenomenon (Cai et al., 2013) also strengthens with dimension, reducing entropy further. Text and multimodal models behave similarly across all tested dimensions.

## E.2  SCALE AND CHUNK ABLATION

Table 7 evaluates how compression varies with batch size and chunk granularity. The transpose and byte-shuffle operations operate across vectors within each compressed unit, so larger batches provide more context for entropy coding. We test two related scenarios: (1) varying total batch size $N$ with full-matrix compression, and (2) fixed $N = 10,000$ with varying chunk sizes for random access.

Compression improves with scale: single-vector compression achieves 1.35×, rising to 1.50× at $N \geq 100$ and plateauing at 1.52× for large batches. Chunking and scale are equivalent: compressing $N = 1$ as a full matrix yields the same 1.35× ratio as chunking 10,000 vectors into single-vector chunks, since both lack cross-vector context for entropy coding. Chunk sizes of 100 to 1,000 achieve practical random access with 1.8 to 2.5% overhead. For a database of 1 million embeddings with chunk size 1,000, retrieving an arbitrary vector requires decompressing at most 1,000 vectors (2.66 ms), with storage overhead under 2%.

The compression loss at small chunks reflects the entropy coder's need for context to build accurate statistical models. An alternative would use a pre-trained arithmetic coder with fixed probability tables from the known angle distributions (Theorem 1), potentially eliminating context dependency. We leave this for future work.

Table 7: Scale and chunking ablation (768d). Top: varying batch size with full-matrix compression. Bottom: fixed $N = 10{,}000$ with varying chunk size for random access.

| Configuration | N | Raw (KB) | Ours (KB) | Ratio | Overhead | Latency |
|---|---|---|---|---|---|---|
| *Scale (full-matrix compression)* | | | | | | |
| N=1 (single vector) | 1 | 3 | 2 | 1.35× | – | – |
| N=10 | 10 | 30 | 23 | 1.32× | – | – |
| N=100 | 100 | 300 | 202 | 1.49× | – | – |
| N=1,000 | 1,000 | 3,000 | 2,003 | 1.50× | – | – |
| N=10,000 | 10,000 | 30,000 | 19,701 | 1.52× | – | – |
| N=100,000 | 100,000 | 300,000 | 197,250 | 1.52× | – | – |
| *Chunking for random access (N=10,000 total)* | | | | | | |
| Chunk=1 (per-vector) | 10,000 | 30,000 | 22,295 | 1.35× | +13.2% | 0.01 ms |
| Chunk=10 | 10,000 | 30,000 | 21,423 | 1.40× | +8.7% | 0.04 ms |
| Chunk=100 | 10,000 | 30,000 | 20,200 | 1.49× | +2.5% | 0.30 ms |
| Chunk=1,000 | 10,000 | 30,000 | 20,052 | 1.50× | +1.8% | 2.66 ms |
| Chunk=10,000 (full) | 10,000 | 30,000 | 19,701 | 1.52× | 0% | 14.8 ms |

## E.3 GEOMETRIC DISTRIBUTION ANALYSIS

Table 8 tests whether compression depends on how vectors are distributed on the sphere, comparing uniform, clustered (von Mises-Fisher with varying $\kappa$), sparse, orthogonal, and real embeddings.

Table 8: Compression across geometric distributions (2000 vectors, 768d). vMF = von Mises-Fisher; $\kappa$ = concentration parameter.

| Distribution | Avg Cos-Sim | Baseline | Spherical | Impr. |
|---|---|---|---|---|
| Uniform on sphere | 0.000 | 1.19× | 1.50× | +20.9% |
| vMF clustered ($\kappa$=50, 5 clusters) | 0.001 | 1.18× | 1.50× | +20.9% |
| vMF moderate ($\kappa$=10) | 0.000 | 1.19× | 1.50× | +20.9% |
| vMF concentrated ($\kappa$=100) | 0.016 | 1.18× | 1.50× | +20.9% |
| vMF tight ($\kappa$=1000) | 0.472 | 1.18× | 1.50× | +21.0% |
| Orthogonal vectors | 0.000 | 1.18× | 1.50× | +20.9% |
| Sparse (10% nonzero) | 0.000 | 5.83× | 7.22× | +19.3% |
| BGE-base (real) | 0.048 | 1.19× | 1.52× | +21.7% |

The improvement holds at 20 to 21% regardless of distribution. Uniform random points, tightly clustered points ($\kappa$=1000, average cosine similarity 0.47), orthogonal vectors, and real embeddings all yield nearly identical compression ratios, confirming that the gain derives from the bounded-angle property of unit-norm vectors, not from inter-vector structure. Sparse vectors already compress well with the baseline due to zero values, but spherical still adds 19%.

## E.4 REDUCED PRECISION FORMATS

We evaluate the spherical transform on BF16, FP16, FP8, and Int8 embeddings. Table 9 compares compression methods across precision formats using 10,000 BGE-base embeddings (768 dimensions).

The spherical transform benefits only float32 embeddings. For all reduced precision formats, baseline outperforms spherical because arccos and arctan2 produce float32 outputs regardless of input precision. When BF16 values are stored in float32 containers, the 16 zero mantissa bits compress well with baseline at 3.10×, but spherical destroys this zero-bit pattern. For two-byte formats like BF16 and FP16, baseline compresses 15.36 MB to 10 to 13 MB, while spherical expands to 20 MB at 0.77× by outputting four-byte values. For one-byte formats, 7.68 MB becomes 20 MB at 0.39×. For reduced precision embeddings, apply baseline directly to the raw bytes.

Table 9: Compression across precision formats (BGE-base, 768d, 10k vectors). Sizes in MB. BF16-as-f32 stores BF16 values in float32 containers with zeroed lower mantissa bits. All other formats use their actual byte width: 4 bytes for float32, 2 bytes for BF16/FP16, 1 byte for FP8/Int8.

| Format | Method | Original | Compressed | Ratio | Max Error | Cosine Error |
|---|---|---|---|---|---|---|
| Float32 | Baseline | 30.72 | 25.27 | 1.22× | 0 | 1.8e-7 |
| Float32 | Spherical | 30.72 | 19.94 | **1.54×** | 6.7e-8 | 1.2e-7 |
| BF16-as-f32 | Baseline | 30.72 | 9.90 | **3.10×** | 2.0e-3 | 2.4e-6 |
| BF16-as-f32 | Spherical | 30.72 | 19.97 | 1.54× | 2.0e-3 | 2.4e-6 |
| BF16 | Baseline | 15.36 | 9.91 | **1.55×** | 9.8e-4 | 2.0e-6 |
| BF16 | Spherical | 15.36 | 19.96 | 0.77× | 9.8e-4 | 2.0e-6 |
| FP16 | Baseline | 15.36 | 12.74 | **1.21×** | 1.2e-4 | 1.2e-7 |
| FP16 | Spherical | 15.36 | 19.96 | 0.77× | 1.2e-4 | 1.8e-7 |
| FP8-E4M3 | Baseline | 7.68 | 5.42 | **1.42×** | 1.6e-2 | 5.6e-4 |
| FP8-E4M3 | Spherical | 7.68 | 19.86 | 0.39× | 1.6e-2 | 5.6e-4 |
| FP8-E5M2 | Baseline | 7.68 | 5.07 | **1.51×** | 3.1e-2 | 2.0e-3 |
| FP8-E5M2 | Spherical | 7.68 | 19.84 | 0.39× | 3.1e-2 | 2.0e-3 |
| Int8 | Baseline | 7.68 | 5.49 | **1.40×** | 1.3e-3 | 2.3e-4 |
| Int8 | Spherical | 7.68 | 19.78 | 0.39× | 1.3e-3 | 2.3e-4 |

