# OpenReview forum: "Embedding Compression via Spherical Coordinates"
_ICLR.cc/2026/Workshop/GRaM — ICLR 2026 Workshop GRaM Poster_

### Official Review · Reviewer_uJ4Y · 2026-02-23
**Strong nearly-lossless embedding compression with impressive gains, but limited generality and practical impact**

**Rating:** 7
**Confidence:** 3

**Review:**

The paper clearly states the context and the objective of the work. It is easy to understand, and the background reminders are helpful for non-expert readers. The figure used to illustrate the method is both simple and effective. The evaluation section is comprehensive, benchmarking the proposed approach against various other methods. Key numerical results are clearly presented, highlighting the significance of the performance gains. The results are impressive; this is one of the best nearly-lossless compression methods in terms of compression rate. Its application to LLMs and other models could significantly reduce storage costs. However, since the impact of embedding compression errors is limited in many applications, lossy methods might still be preferred for even higher compression ratios. Interestingly, I found no other papers proposing this specific method; given its simplicity, it is surprising that it has not been explored previously.

# Pros:
- Improves upon SOTA methods: It outperforms baselines like ZipNN by increasing the compression ratio from 1.2x to 1.5x.
- Efficient Similarity Search: It proposes a similarity algorithm with $O(d)$ complexity, allowing for direct computation in the compressed domain.

# Cons:
- Pseudo-lossless nature: It is not strictly lossless, though the reconstruction error is $<10^{-7}$, which is below the float32 machine epsilon.
- Format limitations: The method is specific to float32 and cannot be easily generalized to other formats like BF16 or INT8 without losing its efficiency.

**Pmlr Suitability:**

Yes

---

### Official Review · Reviewer_sWs2 · 2026-02-24
**Review for submission number 7**

**Rating:** 6
**Confidence:** 3

**Review:**

This paper proposes a training-free, near-lossless compression method for unit-norm embeddings: convert each embedding from Cartesian coordinates to d-1 spherical angles, then apply **transpose + byte-shuffle + entropy coding (zstd)**. The key mechanism is geometric: in high dimensions, spherical angles concentrate near \left(\pi/2\right), which makes IEEE-754 **exponent bytes collapse** (and some mantissa bytes more predictable), enabling better entropy coding.

Empirically, the method achieves about 1.47–1.59 compression across 26 configurations (text/image/multi-vector), improving over a ZipNN-style SOTA baseline. The paper argues reconstruction error is below float32 machine epsilon and therefore preserves retrieval quality, but it does **not** report end-to-end retrieval metrics.

**Strengths**

- **Strong geometry grounding**: explicitly leverages the manifold constraint x\in S^{d-1} (common for cosine-similarity embeddings) and ties compression gains to concentration around $\left(\pi/2\right)$.
- **Clear mechanism + formal statement**: Theorem 1 quantifies exponent collapse: for polar angles with $(d-k)\ge 64$, exponent byte =127 with probability >0.999.
- **Breadth of evaluation** across text/image/ColBERT multi-vector embeddings, all unit-normalized.

**Weaknesses**

- **Missing task-level evaluation**: The claim “preserves retrieval quality” is supported via “error <10^{-7}” reasoning, but there are no retrieval metrics (Recall@k / nDCG / MRR) or ranking-invariance experiments.
- **Novelty is moderate**: components (spherical coordinates, byte shuffling, entropy coding) are known, and the paper itself cites prior spherical-coordinate compression/quantization work (though mostly lossy) and byte-shuffling in HPC.

**Questions**

1. Can you report end-to-end retrieval quality under compression (e.g., Recall@k / nDCG@10 / MRR) for at least one single-vector and one multi-vector setup? Your current argument is purely error-based.
2. Can you elaborate on the tradeoff between d-k and the induced compression error?
3. Can you more crisply position against the two prior-work families you cite: (i) byte-shuffle + entropy coding (Alted/ZipNN) and (ii) spherical-coordinate lossy quantization? What is the conceptual advancement beyond the combination?
4. Can we easily multiply compressed (by your approach) datasets easily, like in quantization-type compression approaches?

**Pmlr Suitability:**

Yes

---

### Meta-Review · Area_Chair_NJnD · 2026-02-25

**Decision:**

Accept

**Metareview:**

In a very literal sense, this paper is within scope for the workshop: it takes advantage of the "geometry" (unit-normedness and concentration of coordinate values) of representations to improve compression of representations. Although short, it is well-presented and presents a compelling empirical view of their method's compression quality. The biggest weakness is that the authors only evaluate the compression quality (quality per compression amount), and not the actual performance on end-to-end retrieval tasks. The authors should please add this comparison prior to publication in the PMLR proceedings.

**Relevance To Proceedings:**

Yes — suitable for PMLR (long paper)

**Relevance To Workshop:**

Yes — suitable for GRaM

---

### Decision · Program_Chairs · 2026-03-02

Accept (Poster)